**PLOS** NEGLECTED TROPICAL DISEASES

# Contribution of epidermal growth factor (EGF) in the treatment of cutaneous leishmaniasis caused by *Leishmania major* in BALB/c mice

Mohammad Saberi[1ʘ], Alireza Latifi[1ʘ], Majid Golkar[2ʘ], Pezhman Fard-Esfahani[3], Sina Mohtasebi[1], Aref Teimouri[4], Mohammad Javad Abbaszadeh Afshar[5], Elham Kazemirad[1‡]*, Mehdi Mohebali[1,6‡]*

**1** Department of Medical Parasitology and Mycology, School of Public Health, Tehran University of Medical Sciences, Tehran, Iran, **2** Department of Parasitology, Pasteur Institute of Iran, Tehran, Iran, **3** Department of Biochemistry, Pasteur Institute of Iran, Tehran, Iran, **4** Department of Parasitology and Mycology, School of Medicine, Shiraz University of Medical Sciences, Shiraz, Iran, **5** Department of Parasitology and Mycology, School of Medicine, Jiroft University of Medical Sciences, Jiroft, Iran, **6** Center for Research of Endemic Parasites of Iran, Tehran University of Medical Sciences, Tehran, Iran

ʘ These authors contributed equally to this work.
‡ EK and MM also contributed equally to this work.
* kazemirad@tums.ac.ir, ekazemirad@yahoo.com (EK); mohebali@tums.ac.ir (MM)

**Data Availability Statement:** All data are in the manuscript and/or supporting information files.

## Abstract

Cutaneous leishmaniasis (CL) is a tropical disease that can cause chronic lesions and leave life-long scars, leading to social stigmatization and psychological disorders. Using growth factors and immunomodulatory agents that could accelerate wound healing and reduce the scar is highly demanded. Epidermal growth factor (EGF) plays an essential role in wound healing. It stimulates the proliferation of keratocytes and fibroblasts, and promotes re-epithelialization. Here, the effect of EGF in combination with Glucantime and nano-liposomal Amphotericin B (SinaAmpholeish) on the healing process of CL in BALB/c mice was investigated. Seventy-two mice were infected with *Leishmania major* parasites and randomly divided into eight treatment groups after the appearance of the lesion. The treatment was continued for five weeks, and lesion sizes were measured weekly. Parasite load was determined in the skin biopsies using qPCR. We found that subcutaneous injection of EGF at 4.5 μg/kg, combined with each of the two antileishmanial drugs, significantly reduced the wound size and parasite load; however, EGF at 1.5 μg/kg failed to be effective. Besides, the wound size and parasite loads were significantly lower in the SinaAmfoleish groups compared to the Glucantime groups. Among the treatment groups, EGF 4.5 μg/kg combined with SinaAmpholeish exhibited the most significant reduction in wound size and parasitic load. Our results suggest that EGF can potentiate the wound healing effect of antileishmanial drugs. Further studies are warranted to explore the beneficial effects of combining EGF with antileishmanial drugs in patients with cutaneous leishmaniasis in order to accelerate wound healing and reduce the scar.

**Funding:** This research was funded and supported by Tehran University of Medical Sciences (TUMS), Tehran, Iran (Grant no.97.03.160.40488 awarded to MM). The funders had no role in study design, data collection and analysis, decision to publish, or preparation of the manuscript.

**Competing interests:** I have read the journal's policy and the authors of this manuscript have the following competing interests: MG and PF-E are founders of Borna Biopharma company providing recombinant human EGF.

## Author summary

Cutaneous leishmaniasis (CL) is a vector-borne disease caused by *Leishmania* that affects more than one million people worldwide each year. CL is a disfiguring disease capable of causing chronic skin ulcers that leave scars with significant social and economic impact. Although chemotherapy, such as antimonial compounds, is the pillar choice for treatment, large doses and extended treatment regimens hamper their efficacy. Application of growth factors and immunomodulatory agents for treatment of CL represents a highly favorable approach for reducing the healing time and scar. Here, we evaluated the effectiveness of subcutaneous administration of epidermal growth factor (EGF) in combination with Glucantime or nano-liposomal Amphotericin B (SinaAmpholeish) in the treatment of CL in BALB/c mice. During the five-week treatment, lesion sizes were measured weekly. At the end of the study, parasite load was determined in the skin biopsies using the qPCR method. Combination therapy of EGF at 4.5 μg/kg with either drug alone enhanced wound healing; however, treatment with EGF and SinaAmpholeish was most potent in reducing the lesion size and parasitic load. Numerous studies indicated the usefulness of EGF for wound healing in different clinical settings. Our results suggest EGF as a promising supplement to antileishmanial drugs for the treatment of CL. Further studies are needed to explore the beneficial effect of combining EGF with antileishmanial drugs in patients with cutaneous leishmaniasis.

## Introduction

Leishmaniasis is a tropical disease caused by the vector-borne protozoan parasite of *Leishmania*. It is a major health problem in four eco-epidemiological regions of the world: the Americas, East Africa, North Africa, and West and SouthEast Asia. According to the World Health Organization (WHO) reports, 99 countries and territories are endemic to leishmaniasis [1,2]. Cutaneous leishmaniasis (CL) is the most frequent clinical presentation, usually represented as a localized lesion, and often leaves disfiguring scars causing psychological, social, and economic problems [1,3]. In 2022, eight countries reported more than 5000 CL cases, including Afghanistan, Algeria, Brazil, Colombia, Iran, Iraq, Peru, and the Syrian Arab Republic, which account for 85% of global estimated CL incidence [2]. In the Old World, the most common etiological agents of CL are *Leishmania major* and *Leishmania tropica* [4,5]. Iran is one of the Middle East's most major endemic countries of CL, where all clinical manifestations and epidemiological aspects of the disease have been reported in recent decades [6].

Concerning the absence of an effective vaccine, chemotherapy still plays a major role in treating and controlling leishmaniasis. Pentavalent antimonials, such as meglumine antimoniate (Glucantime), have been the first-line drugs for treating all clinical forms of leishmaniasis worldwide for over 80 years [7]. However, several disadvantages are associated with this treatment, such as painful and multiple intralesional injections, long-term daily parenteral dosing, high toxicity, and treatment failure [8,9]. The second treatment choice is amphotericin B, which requires multiple injections with numerous side effects, variable efficacy, and drug resistance [8]. Since conventional amphotericin B (AmB) used for intravenous injection has chronic nephrotoxic effects, liposomal formulations of AmB (AmBisome) have been developed to reduce the toxicity and side effects [10,11]. Intravenous (i.v.) injection of AmBisome is mainly effective against visceral leishmaniasis, and the efficacy against cutaneous leishmaniasis is lower due to poor delivery and less accumulation within the lesion [12]. Recently, a topical formulation of nano-liposomal Amphotericin B (SinaAmpholeish 0.2%) was developed and

applied to treat CL caused by *L. major*, resulting in promising outcomes in clinical trial studies [13–15]. The nanosized liposomes as vehicles for AmB contribute to retaining the drug longer at the lesion site, enhancing drug penetration into the dermis where *Leishmania* resides, and promoting drug release [16,17]. Although the results of topical AMB are promising in some Old World *Leishmania* species, there is a clear need to increase the cure rate, accelerate the healing process, and reduce the scar. In addition, other local alternative treatments such as topical Paromomycin, cryotherapy, and thermotherapy have also been applied to treat Old World CL [18].

There is solid evidence that several factors related to the immune system and tissue regeneration promote wound healing [19]. Growth factors are critical in all stages of wound healing, including inflammation, angiogenesis, granulation tissue formation, re-epithelialization, matrix formation, and remodeling. Epidermal growth factor (EGF) is a protein secreted by platelets, macrophages, fibroblasts, and keratinocytes and is essential for wound healing. It stimulates proliferation and migration of fibroblasts and keratinocytes, formation of granulation tissue, contraction of wounds, and collagen deposition [20]. Besides, EGF has an immunomodulatory role in inflamed skin tissue, reducing allergen-induced IL-6 production and Th17 responses in the skin [21]. Previous studies have revealed that EGF signaling promotes acute wound healing, which involves the early recruitment of neutrophils and, ultimately, the re-establishment of the physical barrier [22,23]. Several studies established the effectiveness of exogenous recombinant EGF in treating active and chronic wounds, burns, and diabetic ulcers [24,25]. Also, applying exogenous EGF to wounds can induce rapid re-epithelization, thereby reducing cutaneous scars via suppressing inflammatory reactions [26,27]. Several pieces of evidence revealed that chronic refractory wounds exhibit low levels or activity of EGF. Therefore, the administration of exogenous EGF could compensate for the deficiency of EGF in the wound, prompting wound healing and reducing scar formation [28].

A recent study investigated expression levels of EGF in different stages of cutaneous leishmaniasis and upon the treatment. Interestingly, they observed decreased EGF expression in chronic lesions, treatment failure, and relapsing lesions. On the other hand, EGF expression gradually increased in the final stages of wound healing and when adequate response to chemotherapy was induced [29]. One fundamental challenge in treating CL is accelerating the healing process and preventing complications, unappealing scars, and aesthetic dissatisfaction with the patient. Since EGF could contribute to wound healing and reduce the scar, for the first time, we evaluated the combined application of EGF with Glucantime and topical liposomal amphotericin B in the healing process of skin lesions caused by *Leishmania major* in BALB/c mice.

## Materials and methods

### Ethics statement

All experimental animal procedures were reviewed and approved by the Human and Animal Research Ethics Committee of the School of Public Health, Tehran University of Medical Sciences (ethical code: IR.TUMS.SPH.REC.1397.258). This study was accomplished according to the guidelines of the Specific National Ethics for Biochemical Research issued by the Research and Technology Deputy of the Ministry of Health and Medical Education (MOHME) of Iran (issued 2005).

### Compounds

Roswell Park Memorial Institute (RPMI)-1640 medium, heat-inactivated fetal bovine serum (FBS), and the penicillin-streptomycin solution (10,000 U/mL) were purchased from Sigma-

Aldrich (USA). SinaAmpholeish (0.4% nano-liposomal Amphotericin B) was obtained from the company of Nano Sina Exir (Iran). Glucantime was purchased from Sanofi Pharmaceutical industry company (Franc). Giemsa stain was obtained from Merck chemical company (Frankfort, Germany). SinaAmpholeish (0.4% nano-liposomal amphotericin B) was prepared from Company Nano Sina Exir (Iran) and Glucantime from Sanofi (France). The recombinant human epidermal growth factor (rhEGF) was obtained from the Borna Biopharma (Iran) and dissolved in PBS shortly before use.

## Preparation of parasite

*L. major* strain (MRHO/IR/75/ER) was provided by the Department of Medical Parasitology and Mycology, Tehran University of Medical Sciences, Tehran, Iran. The *Leishmania* parasite was maintained in a highly virulent state through a continuous passage in BALB/c mice. Amastigotes of the *L. major* strain were isolated from the lesion of infected BALB/c mice and were transformed into promastigotes in RPMI supplemented with 10% FBS and 100 U/mL of penicillin and 100 μg/mL of streptomycin sulfate at 25±1˚C. The *Leishmania* promastigotes were subcultured every four days; promastigotes from the stationary phase, between the fourth and fifth days of culture, were used to inoculate BALB/c mice.

## Animals

Experiments with animals were carried out according to the ethical protocol "ARRIVE" guidelines (https://arriveguidelines.org), and measures were taken to protect animals from pain or discomfort [30]. The protocol was approved by the Tehran University of Medical Sciences ethical review board. We investigated the chemotherapy in BALB/c mice, which are genetically highly susceptible to *L. major* due to the early secretion of IL-4 and Th2-type response leading to non-healing progressive lesions [31].

Seventy-two male BALB/c mice, 6–8 weeks old, weighing approximately 20 g, were purchased from the Razi Institute of Iran, Hesarak, Karaj, Iran. All the mice were kept in standard boxes and housed in a controlled animal care facility, including 23±2˚C temperature, 55–60% humidity, and 12 h of light-dark cycles with free access to water and a standard diet. The animals were inoculated by subcutaneous injection of 100 μL PBS containing $2 \times 10^6$ *L. major* promastigotes from the stationary phase into the base of the tail of mice by insulin syringe.

## Treatment of infected mice

Here, we evaluated perilesional administration of 1.5 or 4.5 μg/kg of EGF subcutaneously in combination with Glucantime (20 mg Sb5+/kg/day IM, 21 days) and SinaAmpholeish gel (daily, five weeks) to determine the effectiveness against lesion progression. The treatment regimen of 1.5 μg/kg of EGF was selected for treating CL animals as EGF is used at 1 μg/kg for treating patients with diabetic foot ulcers [32]. We also used EGF at the higher dosage of 4.5 μg/kg since drug metabolism in rodents is faster than in humans. Moreover, human EGF used in this study is presumably less effective in mice than in humans due to the structural difference.

Four weeks post-infection, when lesions were fully developed, mice were selected according to the same lesion size to ensure similar infection levels. Then, the mice were divided into eight groups, nine mice per group, as follows: C: Untreated Control, infected mice without treatment; N: Normal saline Control, infected mice received 200 μL of Normal Saline via subcutaneous (SC) injection every other day for five weeks; G: Glucantime treatment, infected mice treated with 20 mg Sb5+/kg once daily for 21 days via intramuscular (IM) injection; G +E1.5 and G+E4.5: infected mice received Glucantime treatment, (20 mg $Sb^{5+}$/kg/day IM, 21

**Table 1. Description of treatment groups.**

| Groups | Treatment | Dosage and duration of treatment | Dosage and duration of treatment with EGF |
|---|---|---|---|
| C | Control group | Without treatment | - |
| N | Normal Saline | 200 μL, 0.9% SC injections, alternate-day (5 weeks) | - |
| G | Glucantime | 20 mg Sb$^{5+}$/kg / day IM injection (21 days) | - |
| G+E1.5 | Glucantime | 20 mg Sb$^{5+}$/kg / day IM injection (21 days) | 1.5 μg/kg of EGF, SC injection, alternate-day (5 weeks) |
| G+E4.5 | Glucantime | 20 mg Sb$^{5+}$/kg / day IM injection (21 days) | 4.5 μg/kg of EGF, SC injection, alternate-day (5 weeks) |
| S | SinaAmpholeish | 1 g nano-liposomal amphotericin B/day Topical gel (5 weeks) | - |
| S+E1.5 | SinaAmpholeish | 1 g nano-liposomal amphotericin B/day Topical gel (5 weeks) | 1.5 μg/kg of EGF, SC injection, alternate-day (5 weeks) |
| S+E4.5 | SinaAmpholeish | 1 g nano-liposomal amphotericin B/day Topical gel (5 weeks) | 4.5 μg/kg of EGF, SC injection, alternate-day (5 weeks) |

Subcutaneous (SC), Intramuscular (IM)

days), plus EGF 1.5 μg/kg and EGF 4.5 μg/kg (200 μL), respectively, subcutaneous (SC) injection every other day for five weeks; S: SinaAmpholeish gel (0.4% nano-liposomal amphotericin B), infected mice treated topically once a day for five weeks; S+E1.5 and S+E4.5: infected mice received topical SinaAmpholeish gel daily, plus EGF 1.5 μg/kg and EGF 4.5 μg/kg, respectively, subcutaneous (SC) injection every other day for five weeks (Table 1).

Four weeks post-infection, when local lesions were apparent, the treatment scheme was initiated for five weeks. Lesions were washed with Normal Saline, and then 200 μL of rhEGF solution (1.5 or 4.5 μg/kg) was injected subcutaneously on alternate days using a standard disposable syringe with 27G×0.5 needles. The solution was first injected into the dermal-epidermal junction at three equidistant points all over the lesion contours and then down into the wound bottom to ensure uniform distribution. The same volume (200 μL) of Normal Saline (0.9%) was also injected subcutaneously as the injection control group. Regarding antileishmanial drugs, Glucantime was administrated daily via intramuscular injection (IM) for 21 days alone and in combination with EGF, and topical SinaAmpholeish gel was applied once a day for five weeks alone and in combination with EGF.

After appearing the lesion and at the end of the study, samples from each lesion were prepared. Slide smears were fixed with absolute methanol and stained with Giemsa to detect the amastigotes by the light microscope (1000×). The diameter of lesions was measured weekly before and during a five-week treatment by metric caliper.

## Evaluation of parasite burden in treated mice by quantitative real-time PCR

At the end of the fifth week of treatment, three mice from each group were randomly selected and euthanized; further, about 25–50 mg of tissue from the infected base of the tail was obtained [17,33]. Genomic DNA was extracted using a High Pure PCR template preparation kit (Roche, Germany) according to the protocol provided by the manufacturer. Total genomic DNA was quantified in a NanoDrop 2000 spectrophotometer (Thermo Fischer Scientific, USA) and then used to evaluate parasite burden by quantitative real-time PCR. Each qPCR was performed in 20 μl reactions containing 40 ng target DNA, 100 nM forward and reverse primers, and 1x SYBR Premix Ex Taq II (Takara, Tokyo, Japan). Two sets of primers were

used, including forward [JW11], 5′-CCTATTTTACACCAACCCCCAGT-3′ and reverse [JW12], 5′-GGGTAGGGGCGTTC TGCGAAA-3′ that amplify a 116-bp fragment of the minicircle kinetoplast DNA (kDNA) of *L. major*, present in about 10,000 copies per parasite [34]. Experiments were carried out in triplicate using a StepOnePlus Real-Time PCR System (Thermo Fisher Scientific). The PCR condition was as follows: activation at 95˚C for 30 s, amplification at 95˚C for 5 s, and 60˚C for 30 s for 40 cycles. Finally, a melting curve analysis was performed using the following cycling parameters: 60˚C for 30 sec, 5˚C temperature changes to the end temperature of 95˚C. The total genomic DNA of *L. major*, previously isolated from promastigotes, was serially diluted in ten-fold dilutions corresponding to $10^6$ to 1 parasite to generate the standard curve, as described earlier [35]. A standard curve was set by plotting the Ct values against different standards with the known concentrations of the parasite's DNA. The number of parasites in the samples was calculated by interpolating the Ct of samples in the standard curve.

### Histological assessment

Mice were euthanized, and the base of the tail with the lesion was dissected. The tissue was fixed with 4% paraformaldehyde (PFA) for 24 h at 4˚C, dehydrated in ascending ethylic alcohols (30–100%), cleared with 100% xylene, and embedded in 100% paraffin. Afterward, samples were cut with a microtome at 5 to 10 μm thickness, stained with hematoxylin-eosin (H&E), and mounted on a glass slide for microscopic observation. In the end, morphological and pathological changes of the lesion tissue were checked using light microscope images [36].

### Statistical analysis

Statistical analysis was conducted using SPSS software v.21 (IBM Analytics, USA). Statistical significances in the mean lesion size and parasite load within the groups were calculated using a one-way analysis of variance (ANOVA) test and post hoc Tukey's test (GraphPad Prism, USA); P values < 0.05 were considered statistically significant unless otherwise stated.

## Results

### Effect of EGF administration on lesion progression in a murine model of CL

Skin lesions were of similar size before starting the treatment course (P > 0.05). Treatments were conducted for five weeks, and lesion sizes were measured weekly. Mean lesion sizes during the five weeks of treatment regimens were presented in Table 2. The statistical analysis showed no significant difference (P > 0.05) between the lesion sizes of the eight groups at the end of the first week of treatment. From the second week onwards, the mean lesion size in every group receiving antileishmanial drugs was significantly lower than the untreated and Normal Saline groups (P < 0.01) (Fig 1A and Table 2). While the lesion sizes slightly increased in groups receiving antileishmanial drugs until the end of week 4 due to the time course of the antileishmanial response, the increase in lesion size was more evident in the control groups (Fig 1A and Table 2).

From the third week, the mean lesion size was significantly lower in the SinaAmpholeish groups compared to the Glucantime ones. At the end of week 4, the groups receiving EGF at 4.5 μg/kg exhibited lower lesion size than others. Notably, SinaAmpholeish plus EGF 4.5 μg/kg (S+E4.5) group presented the lowest wound size (7.68± 0.63 mm$^2$) (P < 0.001). Also, the lesions were significantly smaller in the Glucantime plus EGF4.5 μg/kg (G+E4.5) group compared to the Glucantime and Glucantime plus EGF 1.5 μg/kg (G+E1.5) groups.

**Table 2. Comparison of mean wound size in study groups during five weeks of treatment.** The mice were divided into eight groups, and nine were allocated to each group.

| Groups | Lesion size (mm$^2$) before the treatment, mean±SD | Lesion size (mm$^2$) during five weeks of the treatments, mean±SD | | | | |
|---|---|---|---|---|---|---|
| | | 1 | 2 | 3 | 4 | 5 |
| S+E4.5 | 7.01 ± 1.14 | 7.36 ± 1.07 | 8.013 ± 0.91 *** | 8.21 ± 1.01*** | 7.68 ± 0.63*** | 6.98 ± 0.78**** |
| S+E1.5 | 7.28 ± 1.32 | 7.73 ± 1.09 | 8.61 ± 0.96*** | 9.46 ± 0.8*** | 9.95 ± 0.87 *** | 9.81 ± 0.88**** |
| S | 7.02 ± 1.94 | 8.04 ± 1.85 | 8.65 ± 1.89*** | 9.18 ± 1.95*** | 10.25 ± 1.09*** | 9.91 ± 0.47**** |
| G+E4.5 | 7.31 ± 1.03 | 8.10 ± 0.75 | 8.32 ± 0.85*** | 9.36 ± 0.87*** | 9.41 ± 1.21*** | 8.78 ± 0.82**** |
| G+E1.5 | 7.85 ± 1.03 | 8.20 ± 0.88 | 8.89 ± 0.74*** | 10.52 ± 0.55*** | 11.83 ± 0.87*** | 11.94 ± 1.04**** |
| G | 7.33 ± 1.84 | 8.10 ± 1.5 | 9.13 ± 1.02** | 10.98 ± 1.19*** | 12.40 ± 0.87*** | 12.11 ± 0.79**** |
| N | 8.11 ± 1.23 | 8.83 ± 0.95 | 10.88 ± 1.08*** | 12.48 ± 1.02 ns | 15.08 ± 1.03 ns | 18.65 ± 0.77ns |
| C | 8.10 ± 1.27 | 8.81 ± 0.7 | 11.29 ± 1.12 ns | 13.56 ± 0.66 | 16.29 ± 0.64 | 19.86 ± 0.62 |
| P value vs. Control | P > 0.05 | P > 0.05 | P < 0.01 | P < 0.001 | P < 0.001 | P < 0.0001 |

* P value statistical analysis compared to the control group

**P < 0.01

***P < 0.001

****P < 0.0001 ns. Non-significant. SD: standard deviation.

**S+E4.5:** SinaAmpholeish gel (daily) with subcutaneous injection EGF 4.5 μg/kg (Alternate day)

**S+E1.5:** SinaAmpholeish gel (daily) with subcutaneous injection EGF 1.5 μg/kg (Alternate day)

**G+E4.5:** Intramuscular injection of Glucantime 20 mg/kg (daily) with subcutaneous injection EGF 4.5 μg/kg (Alternate day)

**G+E1.5:** Intramuscular injection of Glucantime 20 mg/kg (daily) with subcutaneous injection EGF 1.5 μg/kg (Alternate day)

**G:** Intramuscular injection of Glucantime 20 mg/kg (daily)

**S:** SinaAmpholeish gel (daily)

**N:** Subcutaneous injection of Normal Saline (Alternate day)

**C:** Control without drug treatment

After five weeks of treatment, the highest mean lesion size was observed in the untreated control (19.86± 0.62 mm$^2$) and Normal Saline (18.65± 0.77 mm$^2$) groups, whereas lesion size in SinaAmpholeish plus EGF 4.5 μg/kg (6.98± 0.78 mm$^2$) were the smallest among all groups (P < 0.0001). In the group treated with Glucantime plus EGF 4.5 μg/kg, the mean wound size was 8.78± 0.82 mm$^2$, significantly smaller than the Glucantime plus EGF 1.5 μg/kg and Glucantime groups (P < 0.0001) (Fig 1). No significant difference was observed between Glucantime plus EGF 4.5 μg/kg, SinaAmpholeish, and SinaAmpholeish plus EGF 1.5 μg/kg groups (P > 0.05). There was no significant difference in the mean wound size between groups receiving each of the two antileishmania drugs and their combination with EGF 1.5 μg/kg at any given time during the treatment course (P > 0.05). Overall, the SinaAmpholeish treatment groups exhibited a more significant wound size reduction than the Glucantime groups. Administration of EGF 4.5 μg/kg accompanied with SinaAmpholeish or Glucantime significantly reduced the lesion progression compared to monotherapy (Fig 1, Tables 2 and S1).

## Evaluation of parasite load in the treated BALB/c

The parasite load in the BALB/c was assessed using quantitative real-time PCR at the end of week five of treatment. As shown in Fig 2, the *L. major* parasite burden was significantly reduced in all experimental groups compared to the untreated control groups. Administration of EGF 4.5 μg/kg accompanied with SinaAmpholeish or Glucantime significantly reduced the parasite load compared to monotherapy. The group treated with SinaAmpholeish plus EGF 4.5 μg/kg had the lowest parasite load. There was no significant difference between groups administered SinaAmpholeish alone or plus EGF 1.5 μg/kg. In the group that received

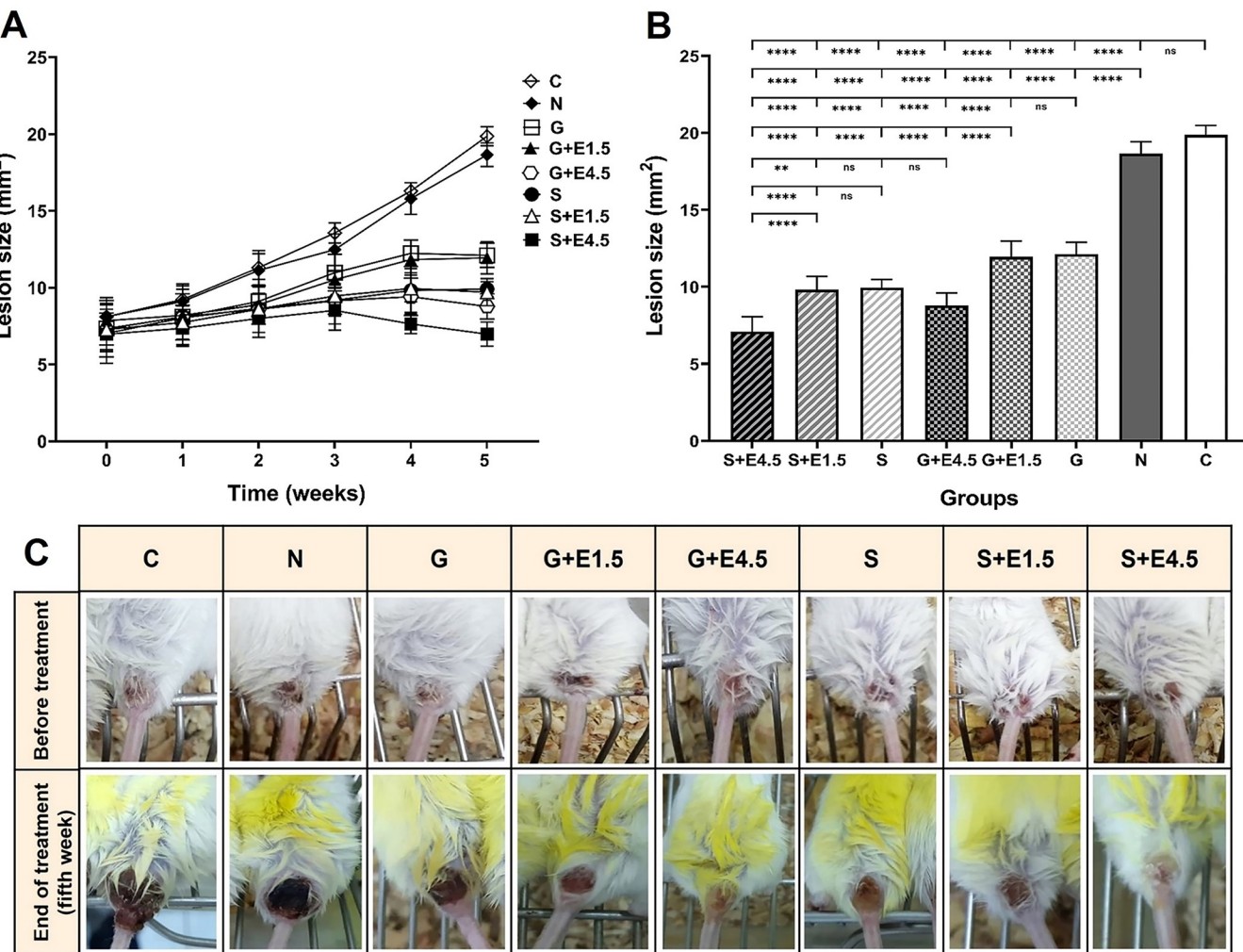

**Fig 1. Efficacy of the combination of EGF (1.5 and 4.5 μg/kg) with Glucantime and SinaAmpholeish treatment against lesion progression in BALB/c mice infected with *Leishmania major*.** Mice were injected (SC) with *L. major* promastigotes at the base of the tail. Animals received a five-week treatment when a nodular lesion had formed at the inoculation site. (A) Evaluation of lesion size progression during five weeks of treatment in groups (n = 9 per group). Each point represents the mean ± SD. (B) Comparison of lesion size in the groups at the end of the fifth week. Statistical analysis was performed with One Way ANOVA, followed by the Tukey post-test, **P < 0.01; ****P < 0.0001. ns: non-significant. (C) Image of the lesions in each group before and at the end of the treatment, photographed by the authors. After week five, the most reduction in lesion size was revealed in groups treated with EGF 4.5 μg/kg with liposomal Amphotericin B (SinaAmpholeish) or Glucantime.

Glucantime combined with EGF 4.5 μg/kg, the parasite load was lower than Glucantome alone or with EGF 1.5 μg/kg. Combination treatment with EGF at 1.5 μg/kg was ineffective in reducing the parasite load, compared to each antileishmanial drug alone (Fig 2 and S2 Table).

## Evaluation of pathologic changes

As shown in Fig 3, severe granulomatous abscess is apparent in the untreated control group. A deep ulcer in the skin and inflammatory cells of the histiocyte/macrophage type with lymphocytic infiltration are seen in the dermis. In foci with chronic lesions, the acute inflammatory phase is seen. Amastigote can also be detected in dead macrophages in the control group where the lesion is apparent. In the group treated with nano liposomal Amphotericin B plus EGF 4.5 μg/kg, the lesion represented the repairing tissue from the edge of the lesion. There is

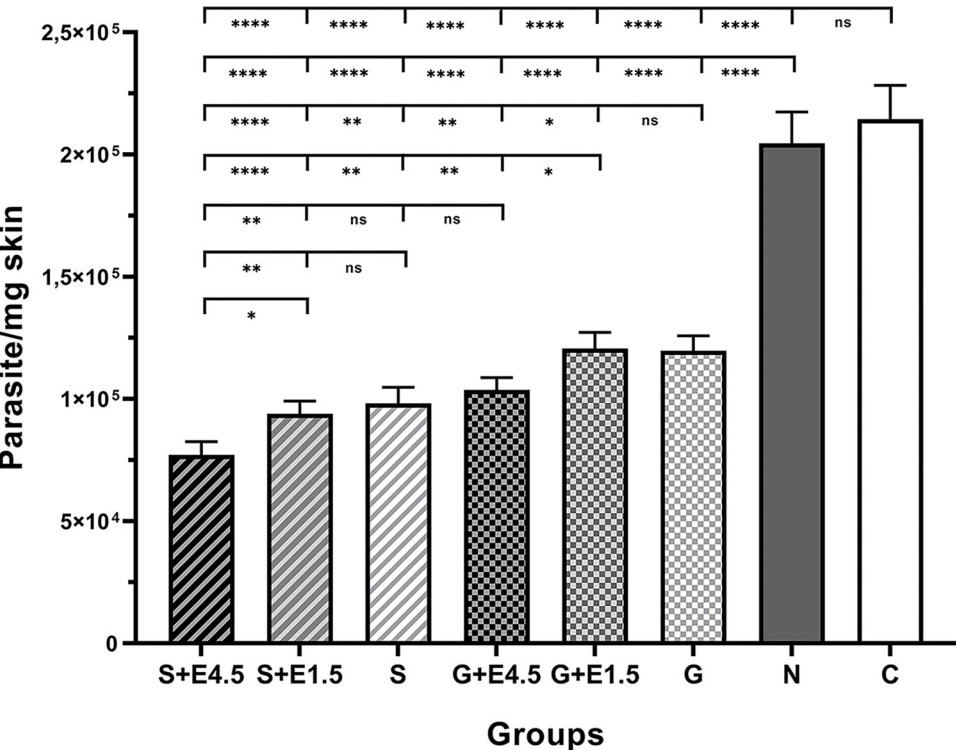

**Fig 2. Parasite burden was determined by quantitative real-time PCR on the skin lesions of animals infected with *L. major*.** At the end of the fifth week of treatment, the DNA was isolated from tissue of the infected base of the tail of the BALB/c. Statistical analysis was performed with One Way ANOVA, followed by the Tukey post-test, *P < 0.05; **P < 0.01; ****P < 0.0001. ns: Non-significant.

almost no trace of the presence of the wound, inflammatory cells, and amastigote bodies, and the wound is repaired.

## Discussion

Cutaneous leishmaniasis is a disfiguring disease that can leave life-long scars and lead to social stigmatization, psychological disorders, and economic losses, especially for women and children [37]. Chemical compounds such as antimonial compounds are the mainstay for the treatment; however, large doses and extended treatment regimens hinder their efficacy. The application of growth factor/immunomodulatory agents that could accelerate the healing time and reduce the scar is highly desirable.

Several growth factors and cytokines regulate different phases of wound healing. In this context, some growth factors showed de-regulated and impaired expression in delayed acute and chronic wounds [38]. Accordingly, several exogenous growth factors, such as EGF, have been used in clinical setting to improve the outcomes of non-healing wounds and reduce scar [25]. EGF signaling is a crucial regulator of epithelial cell proliferation and migration, and increases levels of endogenous hyaluronic acid, collagen, and elastin, thereby influencing the rate of re-epithelialization. EGF has an immunomodulatory role by regulating cytokine and chemokine secretion by keratinocytes and affecting neutrophil recruitment [21,22]. Topical as well as intralesional and perilesional administration of EGF exhibited promising efficacy for accelerating the healing of acute and chronic wounds in humans and animals [39]. Choi et al. demonstrated that EGF treatment in atopic dermatitis (AD) reduced inflammatory reactions

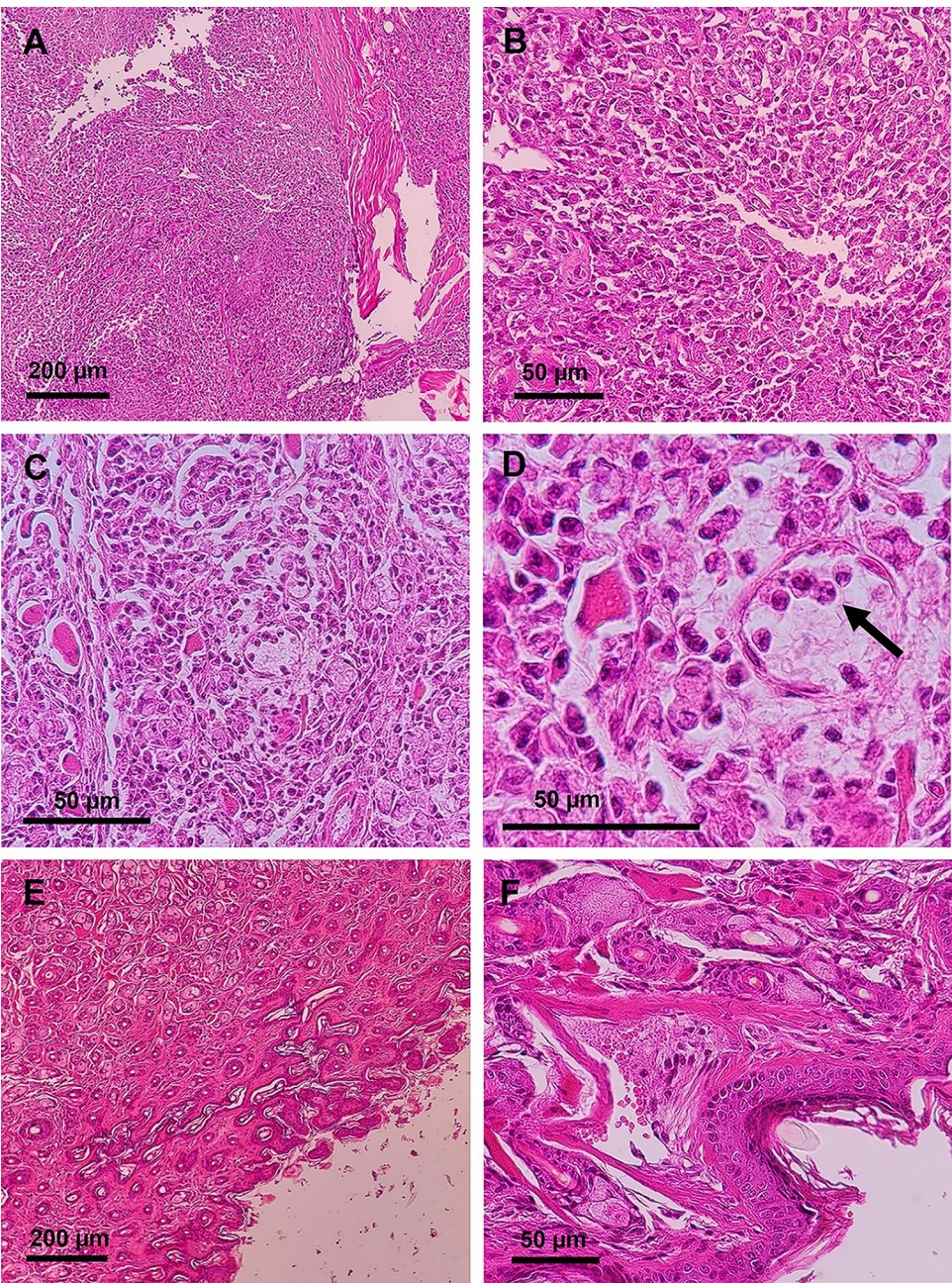

**Fig 3. Representative illustration of pathologic sections of lesions at the end of treatment (week 5). A and B**: Severe granulomatous abscess in pathologic sections of control mice (without treatment). A deep ulcer in the skin and inflammatory cells of the histiocyte/macrophage type with lymphocytic infiltration with intracytoplasmic organisms (pseudo-amastigote) are seen in the dermis; **C and D**: Amastigotes in dead macrophages are illustrated in the untreated control group; arrows indicate amastigotes inside parasitophorous vacuoles; **E and F**: representation of repairing tissue in the lesions treated with nano liposomal Amphotericin plus EGF 4.5 µg/kg. There is a reduction in inflammatory cells without amastigote bodies, representing the start of wound repair.

and further allergic inflammatory responses induced by Staphylococcus aureus colonization. Moreover, they showed that EGF increased the expression of antimicrobial proteins, $\beta$ defensin-2, in human epidermal keratinocytes (HEKs) [23]. EGF reduces cutaneous scars by suppressing inflammatory reactions, decreasing the expression of TGFβ1, and mediating collagen

formation [26]. It is an essential factor in healing chronic wounds as the expression of EGF decreases in the wound formation stage but increases at the final stages of wound healing [24]. In this line, several studies have established the application of topical EGF in accelerating the healing time in chronic wounds, such as venous, arterial, and diabetic ulcers, burns, and infected wounds like leprosy [28].

To date, limited studies have investigated growth factor expression in cutaneous leishmaniasis. It was revealed that in the healing stage of CL in C57BL/6 mice, Th1 cytokines and keratinocyte growth factor (KGF) are critical in pseudo-epitheliomatous hyperplasia development and the normal healing process [40]. Also, the transforming growth factor (TGFβ) effectively stabilized cutaneous leishmaniasis infection in the early stages of wound formation and was upregulated in non-healing lesions [19]. In the healing process of cutaneous leishmaniasis caused by *L. braziliensis*, a higher expression of TGFβ1 was more associated with acute infection and failure or relapses after treatment, and the higher levels of EGF were associated with adequate wound healing [29].

Considering that CL induces chronic, long-lasting lesions that cause life-long scarring, immunomodulating biomaterials such as EGF emerged as a possible complementary treatment option to drive skin tissue regeneration in CL. In this regard, for the first time, we assessed the effect of recombinant EGF in combination with systematic Glucantime or topical nano-liposomal amphotericin B (SinaAmpholeish) for treating CL in BALB/c mice, a susceptible animal model. Our results revealed that chemotherapy significantly prevented lesion progression after the fourth week compared with untreated control; however, the development of lesion size was slightly increasing in all the groups during the first weeks due to the time course of the antileishmanial drug response, which was in agreement with previous studies [33,41]. After a five-week treatment course, the mean wound size and parasite load were significantly lower in the SinaAmfoleish groups compared to the Glucantime ones. One possible explanation for the better therapeutic response of nano-liposomal amphotericin B (0.04%) gel is that the concentration of the drug in the CL infection site is higher in the topical administration of amphotericin B than in the systemic treatment of Glucantime [42]. Also, nano-liposomes enhance drug penetration into the skin and prolong drug retention, leading to long-term sustained release at the lesion site [16, 43]. Besides, we used Glucantime at a dosage of 20 mg/kg for 21 days; however, some studies administered Glucantime at higher doses of 100 mg/kg or for a longer course in animal models to attain complete healing regardless of possible toxicity [41,44].

In the present study, two dosages of 1.5 and 4.5 μg/kg EGF in combination with antileishmanial drugs were evaluated. Our results showed that the combination of EGF at 4.5 μg/kg dose and antileishmanial drugs reduced lesion sizes and parasite load, compared to antileishmanial drugs alone. However, EGF at 1.5 μg/kg dose failed to enhance the efficacy of the drugs. It is noteworthy that the optimum perilesional dose of EGF for treatment of diabetic foot ulcers is 1 μg/kg [32]. We used two doses of 1.5 and 4.5 μg/kg to compensate for faster drug metabolism and lower efficacy of human EGF in mice. In contrast to the higher dose, EGF at 1.5 μg/kg in combination with antileishmanial drugs could not enhance wound healing in mice.

Administration of EGF 4.5 μg/kg plus antileishmanial drugs after the fourth week significantly reduced mean lesion size compared to monotherapy with antileishmanial drugs. The possible reason could be due to the time course of EGF response, as several studies demonstrated that EGF gradually, after the fourth week of injection, induces reepithelization and accelerates wound healing [25,45]. In this line, several clinical trial studies revealed that after intralesional injection of a commercial epidermal growth factor-based formulation (Heberprot-P) every other day in chronic and non-healing diabetic foot ulcers, most patients showed

ulcer size reduction and signs of wound healing after 4–5 weeks of treatment [25,46]. The same phenomenon was observed here, as injection of EGF 4.5 µg/kg plus antileishmanial drug showed the maximum effect in reducing lesions after four weeks of treatment.

Regarding the parasitic load, there was a significant reduction in the treatment groups compared to the untreated control groups using qPCR. The parasite load was remarkably reduced in the groups administered antileishmanial drug plus EGF 4.5 µg/kg. The most significant decrease in the parasite load was found in the SinaAmfoleish plus EGF 4.5 µg/kg group. On the other hand, combination treatment with EGF at 1.5 µg/kg was ineffective in reducing the parasite load. Although low parasite load was identified in groups administered antileishmanial drug plus EGF 4.5 µg/kg, repairing tissue at the edge of the lesion was detected in the pathologic sections. Of note, in the group treated with SinaAmfoleish plus EGF 4.5 µg/kg, at the end of treatment, a decrease in inflammatory cells without amastigote bodies in pathologic section represents the start of wound repair. Moreover, as shown in Fig 3, the appearance of the lesion in the groups that received antileishmanial plus EGF 4.5 µg/kg was smaller, and epithelialization was obviously detected at the edge of the lesion compared to monotherapy with Glucantime or amphotericin B. These findings support that the lesion was in the process of wound healing. The consensus of our results revealed that administration of EGF 4.5 µg/kg inhibited the progression and dissemination of lesions in susceptible BALB/c, resulting in size reduction and lower parasite load than monotherapy.

Although the efficacy of exogenous recombinant EGF in treating leishmaniasis has not been investigated, there is evidence of the effectiveness of other immunomodulatory and growth factors such as Autologous platelet gel (APG), granulocyte and macrophage colony-stimulating factor (GM-CSF) on cure rate of cutaneous leishmaniasis [47,48]. In a recent self-controlled clinical trial, autologous platelet gel (APG) prepared from platelet-rich plasma (PRP) was applied topically in combination with Glucantime in 15 patients, which led to faster complete healing in 66% of lesions and partial healing in 34% of the wounds compared to monotherapy. The inflammatory cells in the pathologic sections were decreased in the group receiving APG supplement, resulting in accelerated wound healing [47]. These findings are in harmony with our results in which the reduction in lesion size and diminishing the inflammatory response in pathologic sections was revealed in the groups receiving EGF 4.5 µg/kg plus the antileishmanial drugs. A study evaluated GM-CSF combined with antileishmanial drugs for treating CL. GM-CSF is a hematopoietic growth factor that can activate macrophages to eliminate *Leishmania* and modulate the immunologic response to the lesion site. A randomized, double-blind, placebo-controlled study in Brazil demonstrated that topical administration of GM-CSF with IV injection of meglumine antimoniate reduced the healing time in CL ulcers caused by *L. braziliensis* [48].

In a study by Montoya et al., the mRNA expression of growth factors was assessed during the healing process following treatment in an animal model of CL caused by *L. braziliensis*. The results showed that in the active form of wound, treatment failure, and wound recurrence, TGFβ1 had the highest expression level. On the other hand, following the establishment of infection and the appearance of an ulcer, the expression level of EGF and platelet growth factor (PDEGF) drastically decreased, and with the onset of treatment and during the wound healing process, their expression gradually increased. Indeed, the inflammatory process for eliminating parasites has a negative regulatory effect on EGF expression by macrophages and fibroblasts in infected and inflamed epidermis tissue, leading to the downregulation of EGF [29]. Moreover, decreased expression of EGF has been reported in chronic wounds such as diabetic foot ulcers. Besides, in chronic wounds, growth factors are trapped by the extracellular matrix or may be excessively degraded by proteases, resulting in impairment or delay in wound healing [28]. Consequently, applying exogenous EGF in infected chronic lesions could compensate

for the impairment of EGF in the wound, thus accelerating wound healing [49]. According to the deficiency of EGF in cutaneous leishmaniasis [29], in the present study, intra- and perilesional administration of exogenous recombinant EGF in combination with the antileishmanial drugs synergistically contributed to accelerating the healing of CL. It is worth mentioning that different nanoparticles loaded with recombinant EGF are available in marketing to protect EGF from enzymatic degradation in the proteolytic wound environment. Also, topical nanomaterials of EGF, such as gel, cream, or spray, were used in several clinical trials, which is more feasible and painless than intralesional or subcutaneous administration [28]. Using the topical nano-based delivery system of EGF plus topical antileishmanial drugs such as nano-liposomal amphotericin B could be a promising tool to accelerate wound healing of CL in the future. Moreover, the combination of EGF with other local treatment regimens like topical Paromomycin or physical treatment regimens such as cryo-or-thermo therapy can also be evaluated in future investigations. Nevertheless, further analysis and clinical trials are needed to introduce recombinant EGF as a complementary to accelerate the treatment of disfiguring CL and decrease scar formation, which could profoundly have a positive effect on the quality of life of patients.

The main limitation of this work was using Glucantime at a dosage of 20 mg $Sb^{5+}$/kg for 21 days. Using a higher dose of Glucantime and extending the treatment course might be required to ensure complete wound healing in the treated groups. The other limitation of this study is that the efficacy of EGF, without combination with antileishmanial drugs, in preventing lesion progression was not evaluated. We did not include a control EGF group since it is not considered an antimicrobial agent [27,28]. In fact, the primary function of EGF is to accelerate wound healing by enhancing re-epithelization. As such, topical application of EGF plus silver sulfadiazine or with silver dressing is extensively used to promote wound healing [49–51]. Recently, autologous platelet gel (APG) containing natural growth factors in combination with Glucantime was administrated in a limited number of CL patients, which led to accelerating wound healing compared to monotherapy with Glucanime [47]. Consistently, our study showed that the simultaneous application of exogenous recombinant EGF and antileishmanial drugs improved the healing of cutaneous leishmaniasis. Nevertheless, more studies are required to investigate the effect of exogenous EGF on the wound healing process, the anti-inflammatory response, and scar formation in cutaneous leishmaniasis.

## Conclusion

We investigated the effect of subcutaneous administration of EGF in combination with antileishmanial drugs for treating CL. Our results demonstrated that the application of EGF at 4.5 μg/kg, in combination with SinaAmfoleish or Glucantime, significantly decreased the lesion size and parasite load. Histopathological findings revealed repairing tissue and reduced inflammatory cells in combination therapy with EGF 4.5 μg/kg. In order to better understand the efficacy and impact of EGF administration on the healing of CL, further studies are required to elucidate expression levels of pro and anti-inflammatory cytokines in the process of wound healing during the treatment. Furthermore, we suggest evaluating topical nano-based EGF in combination with the antileishmanial drugs in the animal model and human clinical trials, which could be a promising new treatment to accelerate wound healing and reduce scar formation.

## Supporting information

**S1 Table. The wound size of BALB/c mice infected with *L. major* in study groups during five weeks of treatment.**
(PDF)

**S2 Table. Parasite burden was determined by quantitative real-time PCR on the skin lesions of BALB/c mice infected with *L. major* at the end of the fifth week of treatment.** (PDF)

## Acknowledgments

The authors would like to acknowledge Borna Biopharma Company for providing recombinant epidermal growth factor. We thank Dr. Ahad Mohammadnejad for helping with the histopathological assessment.

## Author Contributions

**Conceptualization:** Elham Kazemirad.

**Data curation:** Mohammad Saberi, Alireza Latifi.

**Formal analysis:** Alireza Latifi, Elham Kazemirad.

**Funding acquisition:** Mehdi Mohebali.

**Investigation:** Mohammad Saberi, Alireza Latifi, Sina Mohtasebi, Mohammad Javad Abbaszadeh Afshar, Elham Kazemirad.

**Methodology:** Mohammad Saberi, Alireza Latifi, Majid Golkar, Pezhman Fard-Esfahani, Aref Teimouri.

**Project administration:** Mohammad Saberi, Elham Kazemirad, Mehdi Mohebali.

**Resources:** Mehdi Mohebali.

**Supervision:** Elham Kazemirad, Mehdi Mohebali.

**Validation:** Majid Golkar, Mehdi Mohebali.

**Visualization:** Majid Golkar.

**Writing – original draft:** Alireza Latifi, Elham Kazemirad.

**Writing – review & editing:** Majid Golkar, Elham Kazemirad, Mehdi Mohebali.

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
