## [Decision Letter · Decision Letter 0]

22 Jan 2024

Dear Prof. Mohebali,

Thank you very much for submitting your manuscript "Contribution of epidermal growth factor (EGF) in the treatment of cutaneous leishmaniasis caused by Leishmania major in BALB/c mice" for consideration at PLOS Neglected Tropical Diseases. As with all papers reviewed by the journal, your manuscript was reviewed by members of the editorial board and by several independent reviewers. In light of the reviews (below this email), we would like to invite the resubmission of a significantly-revised version that takes into account the reviewers' comments. 

We cannot make any decision about publication until we have seen the revised manuscript and your response to the reviewers' comments. Your revised manuscript is also likely to be sent to reviewers for further evaluation.

Sincerely,

Claudia Ida Brodskyn

Academic Editor

Abhay Satoskar

Section Editor

Reviewer's Responses to Questions

**Key Review Criteria Required for Acceptance?**

**Methods**

-Are the objectives of the study clearly articulated with a clear testable hypothesis stated?

-Is the study design appropriate to address the stated objectives?

-Is the population clearly described and appropriate for the hypothesis being tested?

-Is the sample size sufficient to ensure adequate power to address the hypothesis being tested?

-Were correct statistical analysis used to support conclusions?

-Are there concerns about ethical or regulatory requirements being met?

Reviewer #1: The authors test the effect of EGF in combination with anti-leishmanial therapy on wound healing and parasite load. The sample size used to measure lesion size for the many groups tested is reasonable. However, in the analysis of parasite load only 3 mice per group are included according to the information given in the methods sections. This number is not sufficient.

Moreover, a group only treated with the high dose EFG alone would have been relevant to include since the effect of this growth factor on Leishmania parasite growth is not known and with EFG having effects on both lesion size and potentially parasite load.

The source of EFG should be included in the methods.

No concerns regarding ethics or the statistical method selected.

Reviewer #2: Are the objectives of the study clearly articulated with a clear testable hypothesis stated? Yes.

-Is the study design appropriate to address the stated objectives? Yes.

-Is the population clearly described and appropriate for the hypothesis being tested? Yes.

-Is the sample size sufficient to ensure adequate power to address the hypothesis being tested? Yes

-Were correct statistical analysis used to support conclusions? Yes

-Are there concerns about ethical or regulatory requirements being met? No.

Reviewer #3: - The objective is clear but the methods and the results are not very clear

- Cutaneous leishmaniasis due to L. major is associated with a self-cure rate above 50%–75% at 4–6 months. So it is not clear how this was assessed.

- Any specific reason to euthanize only 3 mice out of 9

**Results**

-Does the analysis presented match the analysis plan?

-Are the results clearly and completely presented?

-Are the figures (Tables, Images) of sufficient quality for clarity?

Reviewer #1: The results are clearly show. Possibly too clear with the same data shown in three different ways.

The text to figure 5 C, D should state which group the samples came from-

Reviewer #2: -Does the analysis presented match the analysis plan? Yes.

-Are the results clearly and completely presented? Yes.

-Are the figures (Tables, Images) of sufficient quality for clarity? Yes.

Reviewer #3: - Why E1.5 and E4.5 alone groups were not created. 

- Why topical paromomyin group was not created which is an established treatment. 

- If 3 mice were already euthanized then what was the need to anesthetize the tail (line 213). 

-Any specific reason to select only 3 mice per group for analysis

- Why intralesional meglumine antimoniate group was not created

- Why lesions were not observed up to 6 months (24 weeks) when almost half of the lesions in L.major show some self-healing.

**Conclusions**

-Are the conclusions supported by the data presented?

-Are the limitations of analysis clearly described?

-Do the authors discuss how these data can be helpful to advance our understanding of the topic under study?

-Is public health relevance addressed?

Reviewer #1: The conclusion would as stated above benefit from the inclusion of an EFG treated group. It would also be high relevant to address the dissemination of L. major, know to occur in Balb/c mice. 

Importantly the effect on parasite load need more data (n=3, as indicated is not sufficient for conclusion) and it would also be relevant to know that the parasites are dead and not incapsulated in a closing wound.

Limitations of the study could be discussed more.

Reviewer #2: -Are the conclusions supported by the data presented? Yes.

-Are the limitations of analysis clearly described? Yes.

-Do the authors discuss how these data can be helpful to advance our understanding of the topic under study? Yes.

-Is public health relevance addressed? Yes.

Reviewer #3: - Line 51- A correction is needed. Amphotericin B or its lipid complexes are not the pillar choice for treatment of CL treatment.

- Generally, if the lesion is less than 4 cm, L. major lesions are self-curing in 4-6 months. In this study, none of the lesions showed any signs of self-cure. On the contrary lesions have increased in size in C and N groups. 

- It is not explained why lesions in G have increased in size despite of receiving the full course of meglumine antimoniate which is an established and very effective CL treatment. 

- Except S+E4.5 and G+E4.5 groups, the other 6 groups have increased in size of lesions week by week, and no explanation is given. It seems the other 6 groups were refractory to the treatment which is difficult to understand.

- Can decrease in size of the lesions in S+E4.5 and G+E4.5 groups be attributed to self-healing? Also had lesions been observed beyond 5 weeks in the remaining 6 groups, lesions might have shown decrease in size?

- It is not clear why lesions increased in size up to the fourth week in G+4.5 and then suddenly decreased in the fifth week. On the contrary in G+E1.5 lesions have continued to increase in size throughout the five weeks' period. 

- Difficult to interpret the results from table

**Editorial and Data Presentation Modifications?**

Reviewer #1: The figures 1-3 show the same thing and should be combined into one figure (1a-c). Also the figure 1 shows the same as table 2 and is repetitive. The table is the more informative.

Reviewer #2: Apart from a few old references that need to be updated and some English terms that need to be improved, the article is well-written and reliably addresses the results discussed.

Reviewer #3: (No Response)

**Summary and General Comments**

Reviewer #1: Therapies aiming at reducing the scaring that can be caused by Leishmania are desirable for many and the scope of this study is relevant.

As is stands the scope of the study is very narrow, with one large experiment performed and with the amount of data presented being limited. More extensive analysis of parasite load, dissemination, tissue responses etc would increase the value of the study. Importantly understandning how EFG alone affects parasite load, dissemination and lesion size is highly relevant.

Reviewer #2: Not applicable.

Reviewer #3: Major revisions and explanations needed

PLOS authors have the option to publish the peer review history of their article (what does this mean?). If published, this will include your full peer review and any attached files.

Reviewer #1: No

Reviewer #2: No

Reviewer #3: Yes: Saurabh Jain
---

## [Decision Letter · Decision Letter 1]

14 Aug 2024

Dear Dr Mohebali,

Thank you very much for submitting your manuscript "Contribution of epidermal growth factor (EGF) in the treatment of cutaneous leishmaniasis caused by Leishmania major in BALB/c mice" for consideration at PLOS Neglected Tropical Diseases. As with all papers reviewed by the journal, your manuscript was reviewed by members of the editorial board and by several independent reviewers. In light of the reviews (below this email), we would like to invite the resubmission of a significantly-revised version that takes into account the reviewers' comments. 

We cannot make any decision about publication until we have seen the revised manuscript and your response to the reviewers' comments. Your revised manuscript is also likely to be sent to reviewers for further evaluation.

Sincerely,

Abhay R Satoskar

Section Editor

Abhay Satoskar

Section Editor

Reviewer's Responses to Questions

**Key Review Criteria Required for Acceptance?**

**Methods**

-Are the objectives of the study clearly articulated with a clear testable hypothesis stated?

-Is the study design appropriate to address the stated objectives?

-Is the population clearly described and appropriate for the hypothesis being tested?

-Is the sample size sufficient to ensure adequate power to address the hypothesis being tested?

-Were correct statistical analysis used to support conclusions?

-Are there concerns about ethical or regulatory requirements being met?

Reviewer #3: There are no issues with the method.

**Results**

-Does the analysis presented match the analysis plan?

-Are the results clearly and completely presented?

-Are the figures (Tables, Images) of sufficient quality for clarity?

Reviewer #3: 1. It is still not clear in the manuscript that why the lesions in the Glucantime treated mice continued to increase in spite of chemotherapy. 

2. Generally 50-75% of CL lesions due to L.major heal spontaneously within 4-6 months. How to ascertain that self-healing didnt take place in any of the groups.

3. Although there is a reduction in the lesion size in the two intervention groups (S+E4.5 and G+E4.5) but complete healing of the lesions is not seen in any of the lesions. Authors need to provide clarification for that. 

4. As per the Manual for case management of cutaneous leishmaniasis in the WHO Eastern Mediterranean

Region, lesions due to L.major can be treated with Cryo-or-Thermo therapy plus intralesional antimonials or topical paromomycin. However, authors have not mentioned or provide clarification of not using these interventions in any of the groups.

**Conclusions**

-Are the conclusions supported by the data presented?

-Are the limitations of analysis clearly described?

-Do the authors discuss how these data can be helpful to advance our understanding of the topic under study?

-Is public health relevance addressed?

Reviewer #3: Major revision is suggested

1. Introduction section- in the first paragraph, there is a need to revise the global situation of CL and number of endemic countries. Authors can refer to WHO's recent publication of Global Leishmaniasis Surveillance Updates 2022 published in WHO weekly epidemiological record.

2. It is still not clear in the manuscript that why the lesions in the Glucantime treated mice continued to increase in spite of chemotherapy. 

3. Generally 50-75% of CL lesions due to L.major heal spontaneously within 4-6 months. How to ascertain that self-healing didnt take place in any of the groups.

4. Although there is a reduction in the lesion size in the two intervention groups (S+E4.5 and G+E4.5) but complete healing of the lesions is not seen in any of the lesions. Authors need to provide clarification for that. 

5. As per the Manual for case management of cutaneous leishmaniasis in the WHO Eastern Mediterranean

Region, lesions due to L.major can be treated with Cryo-or-Thermo therapy plus intralesional antimonials or topical paromomycin. However, authors have not mentioned or provide clarification of not using these interventions in any of the groups.

**Editorial and Data Presentation Modifications?**

Reviewer #3: (No Response)

**Summary and General Comments**

Reviewer #3: (No Response)

PLOS authors have the option to publish the peer review history of their article (what does this mean?). If published, this will include your full peer review and any attached files.

Reviewer #3: Yes: Saurabh Jain
---

## [Decision Letter · Decision Letter 2]

6 Dec 2024

Dear Dr Mohebali,

We are pleased to inform you that your manuscript 'Contribution of epidermal growth factor (EGF) in the treatment of cutaneous leishmaniasis caused by Leishmania major in BALB/c mice' has been provisionally accepted for publication in PLOS Neglected Tropical Diseases.

Best regards,

Abhay R Satoskar

Section Editor

Abhay Satoskar

Section Editor

Shaden Kamhawi

co-Editor-in-Chief

Paul Brindley

co-Editor-in-Chief

Reviewer's Responses to Questions

**Key Review Criteria Required for Acceptance?**

**Methods**

-Are the objectives of the study clearly articulated with a clear testable hypothesis stated?

-Is the study design appropriate to address the stated objectives?

-Is the population clearly described and appropriate for the hypothesis being tested?

-Is the sample size sufficient to ensure adequate power to address the hypothesis being tested?

-Were correct statistical analysis used to support conclusions?

-Are there concerns about ethical or regulatory requirements being met?

Reviewer #3: Ok

**Results**

-Does the analysis presented match the analysis plan?

-Are the results clearly and completely presented?

-Are the figures (Tables, Images) of sufficient quality for clarity?

Reviewer #3: Ok

**Conclusions**

-Are the conclusions supported by the data presented?

-Are the limitations of analysis clearly described?

-Do the authors discuss how these data can be helpful to advance our understanding of the topic under study?

-Is public health relevance addressed?

Reviewer #3: Authors have explained clearly and addressed all previous observations and comments.

**Editorial and Data Presentation Modifications?**

Reviewer #3: (No Response)

**Summary and General Comments**

Reviewer #3: Authors have explained clearly and addressed all previous observations and comments.

PLOS authors have the option to publish the peer review history of their article (what does this mean?). If published, this will include your full peer review and any attached files.

Reviewer #3: **Yes: **Saurabh Jain

---

## [Editor Report · Acceptance letter]

7 Jan 2025

Dear Dr Mohebali,

We are delighted to inform you that your manuscript, "Contribution of epidermal growth factor (EGF) in the treatment of cutaneous leishmaniasis caused by Leishmania major in BALB/c mice," has been formally accepted for publication in PLOS Neglected Tropical Diseases.

Best regards,

Shaden Kamhawi

co-Editor-in-Chief

Paul Brindley

co-Editor-in-Chief
